# Live Cell-Based Semi-Quantitative Stratification Highlights Titre-Dependent Phenotypic Heterogeneity in MOGAD: A Single-Centre Experience

**DOI:** 10.3390/ijms26199615

**Published:** 2025-10-01

**Authors:** Donato Regina, Concetta Domenica Gargano, Tommaso Guerra, Antonio Frigeri, Damiano Paolicelli, Maddalena Ruggieri, Pietro Iaffaldano

**Affiliations:** Department of Translational Biomedicines and Neurosciences (DiBraiN), University of Bari “Aldo Moro”, 70124 Bari, Italy; donatoregina9@gmail.com (D.R.); garganoimma@gmail.com (C.D.G.); guerra.tommaso93@gmail.com (T.G.); antonio.frigeri@uniba.it (A.F.); damiano.paolicelli@uniba.it (D.P.); maddalena.ruggieri@uniba.it (M.R.)

**Keywords:** MOGAD, semi-quantitative fluorescence index, diagnostic biomarker

## Abstract

Myelin oligodendrocyte glycoprotein antibody–associated disease (MOGAD) is an inflammatory demyelinating disorder of the central nervous system characterised by heterogeneous clinical and radiological presentations. Accurate interpretation of serum anti–myelin oligodendrocyte glycoprotein (anti-MOG) antibody titres is critical to improve diagnostic precision and prognostic assessment. This single-centre retrospective study evaluated 19 patients diagnosed with MOGAD in 2023, all of whom were seropositive for anti-MOG IgG, as confirmed by live cell-based assays (CBAs) using full-length human MOG and IgG1-specific secondary antibodies. Antibody quantification combined a ratiometric semi-quantitative fluorescence index with classical endpoint dilution titres, enabling classification into low, medium, and high titre groups. Stratification revealed titre-dependent phenotypic heterogeneity: high-titre patients were older at onset and predominantly presented with optic neuritis, often bilateral, and encephalic involvement, whereas low-titre patients more frequently exhibited spinal cord syndromes, cerebellar or brainstem symptoms, and a higher prevalence of cerebrospinal fluid-restricted oligoclonal bands. Semi-quantitative fluorescence ratios correlated consistently with endpoint titres, and exponential decay analysis demonstrated slower signal loss in high-titre sera, confirming assay reliability. No significant association emerged between titre level and monophasic versus relapsing disease course. Anti-MOG antibody titres could serve not only as a diagnostic biomarker but also to capture clinically relevant immunopathological diversity, supporting a titre-stratified approach to diagnosis and early prognostication. Incorporating semi-quantitative metrics alongside clinical and imaging features may refine the diagnostic algorithm and prevent misclassification of atypical presentations.

## 1. Introduction

Myelin oligodendrocyte glycoprotein antibody–associated disease (MOGAD) is an inflammatory demyelinating disorder of the central nervous system (CNS), increasingly recognised in both paediatric and adult populations [1,2]. It typically presents with acute optic neuritis (ON), transverse myelitis (TM), or acute disseminated encephalomyelitis (ADEM), and may also manifest as cerebral cortical encephalitis or brainstem syndromes [3,4]. Unlike multiple sclerosis or aquaporin-4-IgG neuromyelitis optica spectrum disorder (AQP4-IgG NMOSD), MOGAD can follow either a monophasic or relapsing course and is characterised by distinct radiological and immunopathologic features [5].

A defining biomarker of MOGAD is the presence of serum immunoglobulin G (IgG) autoantibodies targeting conformational epitopes of full-length myelin oligodendrocyte glycoprotein (MOG). Among different diagnostic methods, live cell-based assays (CBAs) using full-length human MOG expressed on the surface of transfected cultured human embryonic kidney 293 (HEK293) cells are considered the gold standard due to their superior specificity and sensitivity [6,7]. These assays are typically quantified via flow cytometry or immunofluorescence microscopy [8].

The 2023 International MOGAD Panel [1] recommends using IgG Fc- or IgG1-specific secondary antibodies to enhance assay specificity, particularly within externally validated in-house setups [9].

Fixed CBAs serve as a viable alternative when live assays are unavailable; however, they demonstrate comparatively reduced sensitivity and specificity. Enzyme-linked immunosorbent assays (ELISAs), by contrast, are not recommended owing to their insufficient diagnostic reliability [10].

Interpretation of CBA results should include both categorical descriptors—such as negative, low-positive, or clearly positive—as well as semi-quantitative parameters, including endpoint titres or flow cytometric ratios. A clearly positive outcome in live CBAs is generally characterised by antibody titres that extend significantly beyond the assay’s threshold. In contrast, fixed CBAs typically consider titres of 1:100 or higher as clearly positive, whereas lower titres, particularly those between 1:10 and just below 1:100, are regarded as low-positive [1,9].

Serum remains the preferred sample for testing, as MOG-IgG detection in cerebrospinal fluid (CSF) demonstrates significantly lower sensitivity. Nonetheless, CSF analysis may be considered in select seronegative cases presenting with strong clinical and radiological features suggestive of MOGAD [1,11]. Low-titre MOG-IgG results have been detected in multiple sclerosis (MS), other neurological disorders, and even healthy individuals. These often represent low-affinity or non-pathogenic binding and should be interpreted with caution, especially outside classical clinical presentations [12,13]. In contrast, clearly high titres are reproducible and strongly associated with typical MOGAD presentations.

Given the heterogeneity of clinical presentations and the complexity of serological testing, precise characterisation of MOG-IgG titres and their correlation with clinical and radiological features at onset may enhance diagnostic accuracy and inform early prognostication in MOGAD. This study aims to evaluate a semi-quantitative method—already adopted in previous studies for different purposes [14]—for detecting serum anti–myelin oligodendrocyte glycoprotein (anti-MOG) antibodies in patients with demyelinating disorders, comparing its diagnostic utility against the established dilutional CBA and evaluating the clinical relevance of antibody titres through a stratified titre-based classification.

## 2. Results and Discussion

### 2.1. Results

A total of 19 patients diagnosed with MOGAD were included in the analysis. The diagnosis of MOGAD was established in all patients according to the international diagnostic criteria proposed by the MOGAD Panel in 2023, as outlined by Banwell et al. [1].

To characterise patterns of MOG-IgGs sera, samples—collected from patients at the moment of symptom onset (all patient were treatment-naïve)—were stratified into high (*n* = 7), medium (*n* = 7), and low (*n* = 5) antibody titre groups, in light of the semi-quantitative fluorescence index classification that we categorised into three titre-based groups: low (0.02–0.33), medium (0.34–0.66), high (0.67–1.00) (Figure 1d). One-phase exponential decay fitting was applied to the fluorescence data from each group. In each graph, the grey curve corresponds to the best-fit one-phase exponential decay model applied to the serial dilution data (Figure 1a–c), and the decay constant (K) was extracted as a quantitative measure of signal loss across dilutions: a higher K value indicates a faster signal decay, while a lower K suggests a more gradual loss (Figure 2).

The resulting K values were inversely associated with the initial antibody titre: 0.0328 ± 0.0442 (K mean ± standard deviation, SD) for high-titre, 0.0429 ± 0.0419 (K mean ± SD) for intermediate-titre, and 0.0935 ± 0.0473 (K mean ± SD) for low-titre samples (Figure 2). These findings indicate that sera with higher antibody concentrations maintain detectable signal over a broader dilution range, reflecting slower signal decay.

The overall mean (SD) age at disease onset was 41.16 ± 18.33 years. A significant difference in age was observed across antibody titre groups, with patients in the high-titre group being older (mean 51.43 ± 18.87 years) compared to those in the medium- (36.14 ± 15.57) and low-titre groups (33.8 ± 14.22) (*p* < 0.0001).

Sex distribution revealed a female predominance across the cohort (73.68%). Male representation was 42.86% in the medium-titre group and 28.57% in the high-titre group. No male patients were present in the low-titre group.

The initial Expanded Disability Status Scale (EDSS) score showed a mean score of 2.24 ± 1.57 across the cohort. Notably, the low-titre group exhibited the highest mean EDSS score at onset 3.0 ± 1.82, suggesting more severe initial neurological involvement.

Regarding the immunological profile, the median disappearance titre, reflecting the number of dilutions at which MOG-IgG remained detectable, was substantially higher in the high-titre group (1:3000 (1:320–1:10240)) compared to the medium- (1:320 (1:40–1:1280)) and low-titre groups (median 1:160 (1:40–1:320)). Similarly, the median semi-quantitative fluorescence-based titre showed a progressive increase across groups: 0.15 (0.05–0.27) (low), 0.49 (0.35–0.65) (medium), and 0.94 (0.67–0.99) (high), confirming the internal consistency between the two titre estimation methods.

The presence of oligoclonal bands (OCBs) in cerebrospinal fluid varied between groups, with the low-titre group demonstrating a higher mean number of OCBs (3.2 ± 4.02) (median 1 (0–11)) relative to the medium-titre (2.71 ± 5.06) (median 1 (0–15)) and high-titre groups (0.5 ± 0.76 across six of the seven patients in the subgroup, due to the fact that one of them did not undergo lumbar puncture) (median 0 (0–2)). This revealed a possible diagnostic and prognostic relevance of this finding, due to the lower frequency of intrathecal IgG synthesis in the high-titre subgroup (16.67% versus 28.57% and 40% in the medium- and low-titre groups, respectively).

No clear trend was observed between anti-MOG titre and disease course classification. The monophasic pattern was most frequent in the medium-titre group (71.43%), followed by the high-titre (57.1%) and low-titre (40%) groups.

The initial clinical phenotype at disease onset varied significantly by antibody titre group. Optic neuritis (ON) was the most frequent presentation in the overall cohort (68.4%) and was more prevalent in patients with higher anti-MOG titres: observed in only 20% of the low-titre group compared to 85.7% in both medium- and high-titre groups (*p* <  0.001). Among ON cases, unilateral involvement was predominant in the low-titre group (100%) but decreased with increasing titre (83.33% in medium, 50% in high), whereas bilateral ON became more frequent at higher titres, reaching 50% in the high-titre group.

Spinal cord syndromes were more common among patients with low anti-MOG titres (60%) and rare in the medium- and high-titre groups (14.3% in both). Among those with spinal involvement, sensory symptoms were universally present, while sphincter dysfunctions were limited to the low-titre group.

Cerebellar symptoms at onset were rare, observed only in one patient from the low-titre group.

Demographic, clinical, and immunological features are summarised in Table 1.

These findings suggest a titre-dependent gradient in clinical presentation: higher titres were associated predominantly with optic nerve involvement, often bilateral, whereas spinal and cerebellar presentations were more frequent in patients with lower antibody titres.

Radiological phenotypes at disease onset, as assessed by magnetic resonance imaging (MRI), varied across anti-MOG titre groups. Optic neuritis (ON) was the most common finding in the overall cohort, observed in 57.9% of patients. The proportion of ON increased with higher titres: 20% in the low-titre group, 57.1% in the medium-titre group, and 85.7% in the high-titre group.

Laterality of ON, as assessed by MRI, differed by titre. Unilateral optic nerve hyperintensity in T2/Fluid-Attenuated Inversion Recovery (FLAIR) sequences was observed in 100% of ON cases in the low-titre group (20% of total), 75% in the medium-titre group (42.9% of total), and 33.3% in the high-titre group (28.57% of total), suggesting a trend toward bilaterality with increasing antibody levels. Bilateral optic nerve hyperintensity in T2/FLAIR was absent in the low-titre group and present in 25% and 66.7% of ON cases in the medium- and high-titre groups, respectively.

Spinal cord involvement was markedly more frequent in the low-titre group (80%), compared to only 14.3% and 28.57% in the medium- and high-titre groups, respectively.

Encephalic lesions, including cortical and subcortical involvement, were more prevalent in higher titre groups, reaching 42.9% in the high-titre group, versus 14.3% in the medium- and 20% in the low-titre group.

Cerebellar or middle cerebellar peduncle lesions were detected in 20% of the low-titre patients, 28.6% of the high-titre ones, and none (0%) in the medium-titre subgroup.

Brainstem involvement appeared more frequent among low-titre patients (40% versus 0% and 14.3% in the medium- and high-titre groups, respectively) (Table 2).

Overall, patients with higher MOG-IgG titres tended to present with anterior visual pathway and encephalic involvement, whereas spinal cord lesions predominated in the low-titre group.

### 2.2. Discussion

Previous studies have demonstrated the reliability of methods used for detecting anti-MOG antibodies in patient sera [15], showing that serial endpoint dilutions yield a reproducible decrease in signal, reflecting a proportional relationship between serum dilution and antibody reactivity [14]. In line with these reports, our findings further support the robustness of this approach.

In addition, this study aimed to highlight the reliability of the semi-quantitative ratiometric method used for detecting anti-MOG antibodies in patient sera, as the endpoint dilutions exhibit a consistent pattern of signal decay, indicating an association between dilution and signal intensity.

Our findings from a single-centre cohort of 19 patients diagnosed with MOGAD according to the international diagnostic criteria proposed by the MOGAD Panel in 2023 [1] highlight a potential stratification of clinical and paraclinical features in MOGAD in relation to the magnitude of anti-MOG IgG titres at disease onset. Patients with higher titres more frequently exhibited optic neuritis [16]—often bilateral, in line with earlier studies [17]—and encephalic involvement, while those with lower titres demonstrated a greater prevalence of spinal cord syndromes and a greater degree of intrathecal IgG synthesis. Despite this phenotypic divergence, no clear association emerged between titre level and monophasic versus relapsing disease course in this cohort, indicating that factors other than antibody titres may be associated with relapse risk, as documented in other reports [18]. These results support the notion that—aligning with prior evidence [19]—anti-MOG antibody titre is not only a diagnostic marker but may also reflect underlying immunopathological heterogeneity within MOGAD. Moreover—in agreement with preceding findings—patients in our study with higher titres of anti-MOG antibodies at presentation may be more likely to exhibit radiological features of typical MOGAD demyelination (e.g., bilateral ON [20]), while also showing fewer CSF-restricted OCBs—consistent with a non-MS-like inflammatory pattern [21]. These trends support the hypothesis—already raised in prior works—that MOGAD represents a distinct immunopathological entity with titre-dependent clinical and paraclinical expression [19,22,23].

The prognostic implications of MOG-IgG titres at disease onset remain under investigation. Some studies suggest that elevated titres are associated with increased disease activity, greater lesion burden on MRI, and a higher likelihood of relapse, particularly when antibody seropositivity persists over time [19,24,25,26,27,28]. However, a standardised threshold for titre stratification and its integration into clinical decision-making is lacking [9].

Current international recommendations, including the 2023 diagnostic criteria for MOGAD, endorse live CBAs using serum as the specimen of choice and emphasise qualitative titre cut-offs to distinguish high from low titres, often using end-point dilutions with thresholds (e.g., ≥1:100) as indicators of diagnostic relevance [1,9]. While effective, these methods are reported to be—as stated in other publications—operator-dependent and labour-intensive, limiting their widespread applicability [29]. Our semi-quantitative ratiometric method addresses these limitations by enabling reproducible, observer-independent quantification of anti-MOG IgG levels, meeting the needs already expressed in earlier studies [29,30]. Moreover, this method also offers potential economic benefits. Unlike traditional end-point dilution methods, which require multiple titrations per patient and result in higher consumable material costs, our method reduces the number of required measurements. This reduction not only lowers the material costs but also minimises operator-dependent errors, thereby increasing both efficiency and reliability. This method correlates fluorescence intensity ratios from antibody binding on transfected HEK293 cells expressing MOG, offering a continuous variable for stratification. Our findings are consistent with the previously expressed need for stratification at both the antibody level and the clinical level, as highlighted in prior investigations [30,31].

The stratification into low (0.02–0.33), medium (0.34–0.66), and high (0.67–1.00) titre groups, based on serum anti-MOG IgG titres, confirms that the decay constant K is a useful quantitative indicator of antibody signal persistence across dilutions and correlates with antibody titre.

This three-tiered classification demonstrated a meaningful association with clinical, radiological, and demographic parameters. Concordant with previous works, patients from our cohort with high titres of serum anti-MOG antibodies exhibited older age at onset compared to those in the medium and low titre groups, suggesting a potential link between age and humoral immunoreactivity against MOG [32,33].

Clinically, high-titre patients more frequently exhibited features typical of MOGAD—including bilateral optic neuritis and longitudinally extensive transverse myelitis—while lower titres were occasionally observed in patients with atypical presentations, in line with evidence reported in previous studies [16,25,28,29,30,32,34].

These findings underscore the importance of incorporating semi-quantitative metrics into diagnostic algorithms, not only to improve diagnostic specificity but also to guide prognosis and therapeutic decisions, a persisting unmet need recognised across the literature [28,35,36].

Importantly, even though fulfilling diagnostic criteria for MOGAD, patients with low anti-MOG IgG titres exhibited clinical and radiological features that were not fully consistent with the classical MOGAD phenotype. Rather than indicating diagnostic uncertainty, these findings underscore the need for cautious interpretation of low-positive titres, as such profiles may resemble those observed in MS or other demyelinating disorders. For instance, some patients presented with CSF-restricted oligoclonal bands—a hallmark of MS but infrequent in MOGAD—and with brain lesions resembling those commonly seen in MS patients, while lacking features more typical of MOGAD, such as bilateral optic neuritis or longitudinally extensive transverse myelitis. While the underlying biology remains unclear, our data are consistent with immunologic heterogeneity that may modulate intrathecal OCBs production in patients with CNS demyelinating disease. These observations align with the 2023 international panel recommendations and other related reports, which emphasise the potential for false-positive or low-affinity reactivity in patients without a classic MOGAD presentation [1,9,35,37].

Echoing prior research, we found that medium-titre patients exhibited a more heterogeneous profile and may represent a clinically distinct subgroup that requires longitudinal monitoring to establish diagnostic clarity [38].

Our findings also complement the 2023 panel recommendations, which acknowledge the diagnostic ambiguity of low-positive results. By contextualising titres within a semiquantitative continuum, our approach provides additional granularity, particularly for patients within the intermediate titre range (0.34–0.66), who often pose a diagnostic challenge. This group may benefit from longitudinal monitoring, as their antibody dynamics may inform disease activity or therapeutic response, in agreement with previous studies that have similarly highlighted the uncertainty of low titres [11] and the need for longitudinal follow-up in clinically heterogeneous subgroups [28].

In addition, this semi-quantitative approach provides a more nuanced framework than dichotomous (positive/negative) or fixed cut-off titre assessments. It allows clinicians to integrate serological data with clinical and radiological findings in a stratified diagnostic algorithm. Particularly for patients in the low-to-intermediate titre ranges, this method facilitates risk stratification and helps avoid overdiagnosis of MOGAD in individuals more likely to have MS or related conditions, addressing the diagnostic challenges highlighted in earlier publications on this topic [28,29,33,37].

Furthermore, our findings stress and underscore the previously proposed possible association between higher antibody titres and greater lesion burden or clinical severity, though this requires confirmation in larger, prospective cohorts [25,26]. As such, the ratiometric assay may also have utility in monitoring disease activity and therapeutic response, addressing needs expressed in recent studies that emphasised the importance of reliable tools for guiding early treatment decisions [39] and for capturing the clinical heterogeneity and long-term course of MOGAD [40].

In summary, the ratiometric approach aligns with traditional titration while providing greater diagnostic precision, enabling clearer differentiation of MOGAD from other demyelinating diseases and guiding individualised therapeutic strategies.

## 3. Materials and Methods

### 3.1. Study Population

This retrospective observational study analysed an Italian single-centre cohort of 19 patients, identified between January and December 2023. Inclusion criteria comprised a first symptom presentation suggestive of central nervous system demyelination and seropositivity for anti-MOG IgG, confirmed via serum testing using a live CBA employing full-length human MOG and IgG1-specific secondary antibodies. All patients underwent serum testing for MOG-IgG using a fluorescence-based live CBA, employing HEK293 cells transfected with full-length human MOG and incubated with patient sera [10]. Detection of bound antibodies was performed with a fluorochrome-conjugated IgG1-specific secondary antibody. Signal quantification and titres were expressed semi-quantitatively both as fluorescence ratios and as endpoint dilution titres, referred to as “disappearance titres” (i.e., the last serum dilution at which fluorescence remained above threshold). This dual modality allowed for both qualitative stratification and more granular comparison of antibody burden.

Patients were categorised into three titre-based groups: low (0.02–0.33), medium (0.34–0.66), and high (0.67–1.00), based on the semi-quantitative fluorescence index. In parallel, endpoint dilution titres (expressed as the reciprocal of the last positive dilution, e.g., 1:320) were recorded and analysed as an independent immunological variable reflecting antibody load. These titres were used to support classification and evaluate potential association with clinical and radiological features.

Demographic and clinical variables were extracted from medical records and included age at disease onset, sex, initial clinical phenotype (e.g., optic neuritis, transverse myelitis, acute disseminated encephalomyelitis), and baseline EDSS score. CSF parameters included white blood cell count, protein concentration, and presence of oligoclonal bands. Radiological data were obtained from initial MRI studies of the brain, spinal cord, and optic nerves, to assess lesion distribution, morphology, and enhancement characteristics. Moreover, disease course was classified as either monophasic or relapsing, based on clinical follow-up. The monophasic pattern was defined as the occurrence of a single demyelinating episode without subsequent clinical relapses during the observation period.

### 3.2. Statistical Analysis

Descriptive statistics were calculated for the entire cohort and for each titre subgroup. The results were expressed as mean  ±  standard deviation (or as median and interquartile range (IQR)) for continuous variables and as absolute frequency and percentage for categorical variables. Comparisons between groups of interest were formally conducted with Student’s *t*-test or the chi-square test, depending on the variable’s nature. Differences in age at disease onset across antibody titre groups were assessed using the Kruskal–Wallis test.

Data from the semi-quantitative analysis on serial dilutions were analysed for one-phase exponential decay fitting, allowing the association of the experimental values with the best-fit exponential model describing signal decay.

To detect the linearity of the dilutions, the decay constant (K) was extracted as a quantitative measure of signal loss. K quantifies the rate of signal decrease under dilutions and serves as an objective parameter to evaluate the proportionality and consistency of the assay response. GraphPad Prism 8.0.2 software was used for exponential decay fitting and K constant analysis. All statistical analyses were performed in Windows SPSS Statistics Version 25 (IBM Corporation, Armonk, NY, USA). A threshold of 0.05 was used for statistical significance.

### 3.3. Cell Culture and Transfection

Serum MOG-IgG antibodies were quantified using a validated in-house live CBA, which remains the gold standard for the detection of human MOG-IgG, as recommended by international diagnostic criteria for MOGAD [14]. This method involves transfection of mammalian cells to express full-length human MOG on the cell surface, allowing direct visualisation of antibody binding. Compared to fixed CBAs, ELISA, or immunoprecipitation assays (IA), live CBA provides significantly higher sensitivity and specificity by preserving conformational epitopes of the native MOG protein [12,13].

HEK293 cells were stably transfected with the full-length hMOG-α1 splice variant complementary DNA (cDNA) (FL-MOG-α1) cloned into pCMV6-AC-GFP plasmid and used as the expression substrate in CBA.

HEK293 cells were grown in Dulbecco’s high-glucose medium with stable glutamine (REF 25030081, Invitrogen, Milan, Italy) added with 10% foetal bovine serum (REF A5256701, Invitrogen, Italy) and 100 U/mL penicillin/100 μg streptomycin (REF 15070063, Invitrogen, Italy). Transfections were performed using Lipofectamine 2000 transfection reagent (REF 12566014, Invitrogen, Italy) following the manufacturer’s protocol. Stable clones were maintained in medium with 0.4 mg/mL geneticin (REF 10131027, Invitrogen, Waltham, MA, USA).

### 3.4. MOG Antibody Detection and Semi-Quantitative Analysis for Endpoint Titration

Human MOG-IgG binding was assessed using a live CBA, as previously described [23]. In brief, transfected cells cultured on poly-L-lysine-coated (REF P2636, Sigma-Aldrich, Milan, Italy) 12 mm glass coverslips were incubated with serum samples. Surface-bound MOG-IgG was detected using a goat anti-human secondary antibody conjugated to Alexa Fluor 568 (REF A21090, Invitrogen, Waltham, MA, USA). After staining, the coverslips were mounted onto slides using a fluorescent mounting medium containing DAPI and visualised with a DMRXA fluorescence microscope (Leica Microsystems Srl, Milan, Italy) equipped with a DFC700T colour camera.

Antibody titres were quantified using a ratiometric method (Leica Application Suite, LASX) previously described by Bollo et al. [14], based on cells expressing the fluorescent fusion protein MOG-GFP. Shortly, two microscopy fields were randomly selected using a 20× objective. Within each field, four distinct regions containing two to four cells each were analysed. Fluorescence intensity in each region was measured in grayscale, and background-subtracted values were used to calculate the red/green fluorescence ratio, representing MOG antibody binding. These values ranged from 0 (no staining) to 1 (maximum antibody binding) and were defined as the MOG quantitative ratio (MOGqr).

Endpoint titrations were assessed on positive samples starting at 1:10, with 1:2 dilution steps. The endpoint titre was considered as the last dilution showing fluorescence on the cell surface above threshold, while the value of the semi-quantitative analysis was below the positive threshold value.

## 4. Conclusions

In conclusion, although external validation in multicentre cohorts is still needed to ensure clinical applicability, assess performance across subgroups, and enable prospective replication where feasible, the semi-quantitative ratiometric method not only correlates with the classical dilutional titre but also enhances diagnostic resolution by distinguishing true MOGAD from mimicking conditions. Our findings are concordant with previous studies, while extending their implications by strengthening the evidence base and addressing unmet needs previously highlighted in the literature [14,41,42,43]. Hence, incorporating titre stratification into clinical practice—especially alongside imaging and CSF biomarkers—may significantly improve the accuracy of MOGAD diagnosis and support personalised treatment approaches.

## Figures and Tables

**Figure 1 ijms-26-09615-f001:**
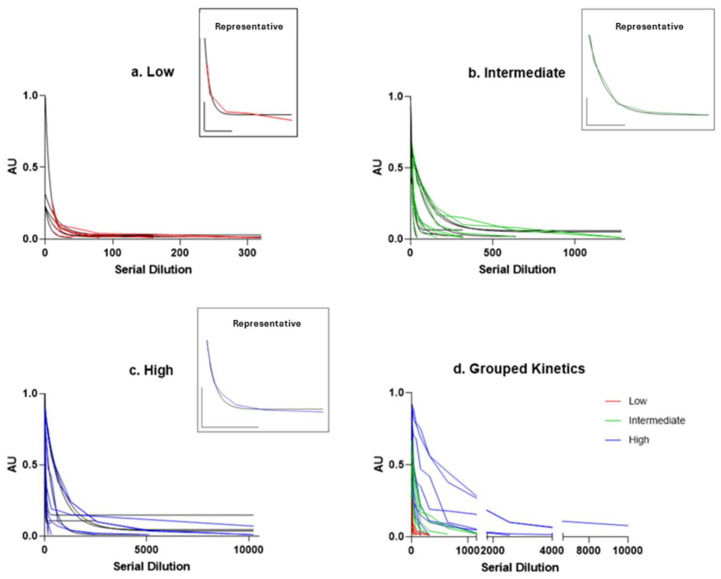
Fluorescence intensity decay. Fluorescence values, expressed in arbitrary units (AU), are plotted against serial dilutions, starting from the highest semi-quantitative value to the disappearance titres. (**a**) Low titre (red), (**b**) intermediate titre (green), and (**c**) high titre (blue) fluorescence decay curves presenting an inset in each panel showing representative kinetics, the corresponding scale bar, and the best-fitted curve. Grey lines represent the ideal regression curve. (**d**) Grouped kinetics of serial dilutions indicated by the colour legend (red, green, and blue). The x-axis has been adjusted to focus on the initial part of the graph, making the kinetics more clearly visible. This fluorescence analysis demonstrates that the semi-quantitative method we employed yields results comparable to those obtained through serial dilutions, supporting its reliability in distinguishing between different disease states.

**Figure 2 ijms-26-09615-f002:**
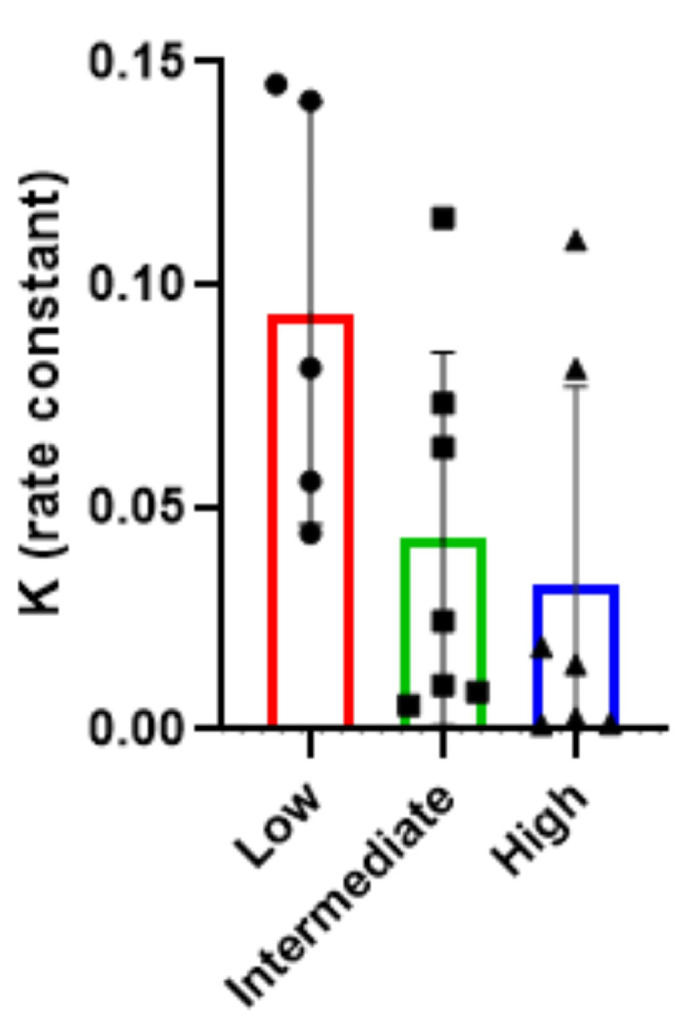
Decay constant (K) values obtained from exponential one-phase decay fitting applied to serial dilution curves of sera stratified by antibody titre. Dots represent individual values, and bars represent the mean K values for low (red, circles) (0.0935 ± 0.0473), intermediate (green, squares) (0.0429 ± 0.0419), and high (blue, triangles) (0.0328 ± 0.0442) titre groups. Error bars indicate the standard deviation.

**Table 1 ijms-26-09615-t001:** Demographic, clinical, and immunological features of the entire cohort and stratified by anti-MOG antibody titres. M: Male; F: Female; EDSS: Expanded disability status scale; Anti-MOG: Anti–myelin oligodendrocyte glycoprotein; CSF: Cerebrospinal fluid; OCBs: Oligoclonal bands; Pts: Patients; ON: Optic neuritis; SD: Standard deviation; tot: total.

	Total	Low Titre (0.02–0.33)	Medium Titre (0.34–0.66)	High Titre (0.67–1.00)
SAMPLE SIZE	19	5	7	7
AGE OF ONSET				
Mean ± SD	41.16 ± 18.33	33.8 ± 14.22	36.14 ± 15.57	51.43 ± 18.87
Median (min-max)	43 (10–70)	30 (17–55)	34 (15–60)	58 (10–70)
SEX DISTRIBUTION				
M (% tot)	5 (26.32%)	0 (0%)	3 (42.86%)	2 (28.57%)
F (% tot)	14 (73.68%)	5 (100%)	4 (57.14%)	5 (71.43%)
CLINICAL FEATURES				
EDSS				
Mean ± SD	2.24 ± 1.57	3.0 ± 1.82	1.64 ± 1.46	2.36 ± 1.22
Median (Min-Max)	2.0 (0.0–6.5)	2.0 (1.5–6.5)	1.5 (0.0–4.0)	2.0 (0.0–4.0)
MONOPHASIC COURSE (% tot)	11 (57.89%)	2 (40%)	5 (71.43%)	4 (57.14%)
ANTI-MOG ANTIBODIES FEATURES				
DISAPPEARANCE TITRE (NUMBER OF DILUTIONS)				
Mean ± SD	2004.21 ± 3092.02	184 ± 117.58	668.57 ± 536.96	4640 ± 3814.98
Median (Min-Max)	320 (40–10240)	160 (40–320)	320 (40–1280)	3000 (320–10240)
SEMI-QUANTITATIVE TITRE				
Mean ± SD	0.56 ± 0.31	0.17 ± 0.09	0.50 ± 0.11	0.90 ± 0.11
Median (Min-Max)	0.57 (0.05–0.99)	0.15 (0.05–0.27)	0.49 (0.35–0.65)	0.94 (0.67–0.99)
CSF FEATURES				
OCBs NUMBER				
Mean ± SD	2.11 ± 4.00	3.2 ± 4.02	2.71 ± 5.06	0.5 ± 0.76
Median (Min-Max)	1 (0–15)	1 (0–11)	1 (0–15)	0 (0–2)
Pts with CSF-RESTRICTED OCBs (% tot)	5 (26.32%)	2 (40%)	2 (28.57%)	1 (16.67%)
CLINICAL FEATURES—ONSET SYMPTOM				
ON (% tot)	13 (68.42%)	1 (20%)	6 (85.71%)	6 (85.71%)
Unilateral ON (% tot) [% ON]	9 (47.37%) [64.29%]	1 (20%) [100%]	5 (71.43%) [83.33%]	3 (42.86%) [50%]
Bilateral ON (% tot) [% ON]	4 (21.05%) [30.77%]	0 (0%) [0%]	1 (14.29%) [16.67%]	3 (42.86%) [50%]
SPINAL CORD (% tot)	5 (26.32%)	3 (60%)	1 (14.29%)	1 (14.29%)
Motor (% tot) [% Spinal Cord]	2 (10.53%) [40%]	1 (20%) [33.33%]	1 (14.29%) [100%]	0 (0%) [0%]
Sensitive (% tot) [% Spinal Cord]	5 (26.32%) [100%]	3 (60%) [100%]	1 (14.29%) [100%]	1 (14.29%) [100%]
Sphincteric (% tot) [% Spinal Cord]	1 (5.26%) [20%]	1 (20%) [33.33%]	0 (0%) [0%]	0 (0%) [0%]
CEREBELLAR (% tot)	1 (5.26%)	1 (20%)	0 (0%)	0 (0%)
BRAINSTEM (% tot)	1 (5.26%)	0 (0%)	0 (0%)	1 (14.29%)

**Table 2 ijms-26-09615-t002:** Radiological features (as assessed by MRI) of the entire cohort and stratified by anti-MOG antibody titres. MRI: Magnetic resonance imaging; ON: Optic neuritis; MCP: Middle cerebellar peduncles; tot: total.

	Total	Low Titre (0.02–0.33)	Medium Titre (0.34–0.66)	High Titre (0.67–1.00)
MRI FEATURES				
ON (% tot)	11 (57.89%)	1 (20%)	4 (57.14%)	6 (85.71%)
Unilateral ON (% tot) [% ON]	6 (31.58%) [54.55%]	1 (20%) [100%]	3 (42.86%) [75%]	2 (28.57%) [33.33%]
Bilateral ON (% tot) [% ON]	5 (26.32%) [45.45%]	0 (0%) [0%]	1 (14.29%) [25%]	4 (57.14%) [66.67%]
ENCEPHALIC (% tot)	5 (26.32%)	1 (20%)	1 (14.29%)	3 (42.86%)
CEREBELLAR—MCP (% tot)	4 (21.05%)	1 (20%)	0 (0%)	2 (28.57%)
BRAINSTEM (% tot)	4 (21.05%)	2 (40%)	0 (0%)	1 (14.29%)
SPINAL CORD (% tot)	7 (36.85%)	4 (80%)	1 (14.29%)	2 (28.57%)
Cervical (% tot) [% Spinal Cord]	5 (26.32%) [71.43%]	4 (80%) [100%]	1 (14.29%) [100%]	1 (14.29%) [50%]
Dorsal (% tot) [% Spinal Cord]	4 (21.05%) [51.14%]	3 (60%) [75%]	0 (0%) [0%]	1 (14.29%) [50%]
Conus (% tot) [% Spinal Cord]	1 (5.26%) [14.29%]	1 (20%) [25%]	0 (0%) [0%]	0 (0%) [0%]
NEGATIVE (% tot)	2 (10.53%)	0 (0%)	2 (28.57%)	0 (0%)

## Data Availability

Data will be shared upon request from qualified researchers.

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
