# Peer review of "Live Cell-Based Semi-Quantitative Stratification Highlights Titre-Dependent Phenotypic Heterogeneity in MOGAD: A Single-Centre Experience"

_ijms, 2025, doi:10.3390/ijms26199615_

Round 1
Reviewer 1 Report
Comments and Suggestions for Authors
Overall, manuscript is well written. No major issues.
- Plots in figures 1 and 2 are hard to make out the curves. Suggest replotting. Also, would a log-scale Y-axis apply here?
- Indentations needed for each new paragraph
- Unclear where the data came from presented in Tables 1 and 2. Also, is all of this information critical or could these tables be reduced.
Reviewer 2 Report
Comments and Suggestions for Authors
Your paper provides interesting and helpful information on titre stratification of CBA and phenotypic presentation in MOGAD. (1) Your findings also raise the question (with an elusive answer for certain) of possible diverse immunologic mechanisms (or the intensity of such) influencing the quantity of intrathecal production of oligoclonal bands in CSF, associated to the level of serological titres. The total % restricted CSF OCBs in your series, nevertheless it is actually in concordance with the reported proportion for MOGAD.
(2) You make a convincing point on the advantages of semiquantitative ratiometric method for serum titres comparison to operator-dependent labor-intensive end-point dilutions. The economic advantage if any, is not discussed. The financial aspect has global interest and implications.
Reviewer 3 Report
Comments and Suggestions for Authors
Given the clinical diversity observed in MOGAD and the many unknowns regarding its mechanisms, examining the relationship between antibody titers and clinical subtypes is important. This paper presents a single-center study of 19 cases, categorizing antibody titers into three groups using their original assay and clarifying the clinical characteristics of each group. These are interesting and valuable findings.
Separating the Results and Discussion sections would improve readability.
The significance of this study is not completely clear for the following reasons and requires revision.
・Why were the subjects divided into three groups based on antibody titers? What is the rationale to make 3 groups, instead of 2 or 4?
・In the Introduction and Discussion sections, previous reports on MOGAD are cited and introduced. The relationship between the content of each sentence and the cited references is in general unclear. Specifically, it is difficult to discern which parts of the text are based on the cited references and which represent the authors' opinions based on the current study. At this moment,it is difficult to judge the points of agreement, disagreement, or novelty between the current findings and the previous reports.
A comprehensive revision is needed for the sentences citing references starting from Reference 9.
Reviewer 4 Report
Comments and Suggestions for Authors
The manuscript focuses on a cohort of MOGAD patients, their stratification, and diagnostic methods.
Specific comments:
1. Consider fixing abbreviations, i.e., introduce them one time and consistently. CBA is defined as "cell-based assays" (line 48) and then as "cell-based antibody assay" (line 59). MOGAD is introduced differently (line 38 and line 82). Lines 47-48 use the abbreviation MOG, but was it introduced in the main text? etc.
2. Figure 1. Red-green-blue colors are explained in the legend; it would be beneficial to also explain them on the Figure itself. Moreover, consider changing the X-axis resolution, as all red are likely below 500, all green are below 2000, and only blue lines go beyond. Thus, 0 to 2000 is the most informative area to illustrate the effect.
3. Moreover, the description of Figure 1 (legend) only focuses on technical aspects, without reference to the text and the subject of the paper. In addition to the technical description of fluorescence and values and colors, consider adding some sentences on the reason it is important for this manuscript.
4. Figure 2, it seems that red-green-blue from Figure 1 are split into Figure 2abc panels, which is redundant. In this light, Figures 1 and 2 can be, or should be, combined. Figure 2 adds very little to Figure 1. Moreover, the description of Figure 2 (legend) is again very technical, and has no clear reference to the text and subject of the paper. Consider illustrating in the Figure legend why this fluorescence analysis is needed for the manuscript and topic described. Moreover, axis Y for the Figure legend can include a range from 0.0 to 1.0; it is unclear why the range should go beyond 1.0 towards 1.5
5. Figure 3. It is suggested to use the dots+columns graph type as more informative.
6. The number style used on Table 1 is not consistent, e.g., "0,67" vs "41.16", and "0,021 vs 0,33"
7. Table 1 has multiple abbreviations. They have to be avoided or explained in the Table legend.
8. Table 2, the same comments as for Table 1 (#6 and #7).
9. It is OK to separate the Results and Discussion sections, as the last part of this combined section is the Discussion. This is optional, of course.
10. It would be nice to have a "Conclusions" section as the very last paragraph of the Discussion, or as a separate one-paragraph small section immediately after the Discussion. Currently, the Conclusion section is present, although it is very far from the Results and Discussions. That is why Materials and Methods go before Results and Discussions.
11. Section 3.3. Materials and Methods. A description of each commercial reagent is expected, including catalog numbers and producers. One gets the impression that everything was purchased in Invitrogen (Italy), although specific catalog numbers are missing, and this is usually needed for reproducibility of the methods.
12. The same for section 3.4. Catalog numbers are needed for the commercial reagents
Reviewer 5 Report
Comments and Suggestions for Authors
This article is a clear and appropriately structured scientific work, fully consistent with the scope of the journal. Despite the existence of prior studies in this field, the present manuscript is timely and of considerable interest for researchers in real clinical practice (clinical neurology, laboratory diagnostics), as well as for specialists in fundamental research areas.
The manuscript addresses the important problem of diagnosis and stratification of patients with MOGAD and presents the results of a retrospective study of a small single-center cohort (19 patients). The text is well-structured, with a logical progression from the introduction and justification of methods to results, interpretation, and conclusion. The reference list is relatively small (31 sources), but it covers both fundamental works and the most recent publications of the past five years, including recommendations of the International MOGAD Panel and several clinical studies. Self-citations are minimal and justified, as they build on the authors’ prior methodological work on semi-quantitative analysis. Overall, the bibliographic base appears relevant and appropriate. The experimental design is aligned with the stated hypothesis: the authors stratified patients into three groups with different antibody levels and demonstrated titre-dependent clinical and radiological differences. Methods are described in sufficient detail, including cell lines, transfection protocols, fluorescence analysis parameters, and statistical tests, ensuring reproducibility by other laboratories. Nonetheless, the sample size is very limited, reducing statistical power (a study limitation), though the authors recognize this and interpret their conclusions cautiously. The graphical material is presented appropriately: the figures clearly illustrate fluorescence dynamics, exponential signal decay, and group differences. Tables concisely summarize clinical, immunological, and MRI data, supporting easy interpretation. Statistical analysis is adequate, and data are processed consistently without methodological contradictions. The conclusions are consistent with the presented results: the authors convincingly demonstrate that anti-MOG antibody titres may serve not only as a diagnostic marker but also as an indicator of clinical and immunopathological heterogeneity. The need for careful interpretation of low titres is emphasized, as well as the appropriateness of incorporating semi-quantitative metrics into the diagnostic algorithm.
Overall assessment of the article.
The article demonstrates novelty and relevance: the research question is clearly and originally formulated, as the authors propose a semi-quantitative stratification of MOGAD patients by antibody titres and demonstrate its clinical and prognostic value. The results are correctly and consistently interpreted, and the conclusions are fully supported by the data. The hypothesis is clearly identified and convincingly confirmed. The findings highlight the role of anti-MOG antibody titres not only as a diagnostic marker but also as an indicator of clinical heterogeneity, which enhances the practical value for clinical practice. The article is written in an academically sound style, with well-structured presentation, figures and tables properly formatted, and analysis performed correctly. The design is appropriate to the aims, methods are described in sufficient detail for reproducibility, and the data appear reliable, internally consistent, and aligned with the study objectives. Collectively, this work can be considered a high-quality, methodologically rigorous study that merits publication.
Thus, the article constitutes a significant contribution to the understanding of the pathogenesis and diagnosis of MOGAD, with particular emphasis on its applied value. The study is conducted at a high scientific level and may be recommended for publication after minor editorial revisions.
Comments/questions:
- It is suggested to clarify and briefly describe how the authors controlled for the possible influence of prior therapies on antibody levels at the time of sample collection, as this may affect the results.
- Please indicate (for example, briefly in the conclusion) how the proposed stratification will be validated in larger cohorts, including multicenter studies, to confirm its clinical applicability.
- It is recommended to expand the reference list (by at least 10+ sources), which would allow for a deeper and more comprehensive discussion of the research problem in the "Results and Discussion" section.
Round 2
Reviewer 3 Report
Comments and Suggestions for Authors
The manuscript has been improved.